# Comprehensive Utilization of Seawater in China: A Description of the Present Situation, Restrictive Factors and Potential Countermeasures

**Shuxin Gong** [1,2], **Hongrui Wang** [1,2,*] , **Zhongfan Zhu** [1,2] , **Qijie Bai** [1,2] **and Cheng Wang** [3]

[1] College of Water Science, Beijing Normal University, Beijing 100875, China; Gongshuxin1995@163.com (S.G.); zhuzhongfan1985@bnu.edu.cn (Z.Z.); 201521470001@bnu.edu.cn (Q.B.)

[2] Beijing Key Laboratory of Urban Hydrological Cycle and Sponge City Technology, Beijing 100875, China

[3] Environmental Science Division, Argonne National Laboratory, Lemont, IL 60439, USA; wangcheng@anl.gov

[*] Correspondence: henrywang@bnu.edu.cn; Tel.: +86-010-58807941

**Abstract:** China is a maritime power. However, as its economy develops rapidly, China lacks freshwater resources. The water resources per capita are low in China and are less than a quarter of the global mean value. The development and utilization of desalination, a new unconventional water resource in coastal areas, has gradually attracted the attention of the central and local governments. This paper introduces three aspects of the comprehensive utilization of seawater in China, including the desalination of seawater, the direct use of seawater, and the use of seawater as a chemical resource. Based on the recent status (2017) of comprehensive seawater utilization in China, the prospects for optimizing the utilization of seawater resources are presented. Furthermore, the restrictive factors and potential countermeasures for the increased use of seawater desalination are investigated. Several recommendations are presented, specifically, improving the laws, using regulations and standards related to desalination, strengthening the policies that support enterprises that use desalination, gradually improving the localization rate of key technologies and equipment, and devoting additional attention to the problems associated with brine processing. Seawater is expected to become an important supplemental source of water in coastal areas of China, and the resources needed for its use will be developed as a strategic and influential industry.

**Keywords:** desalination; direct use of seawater; chemical resource; industrial cycle; countermeasures; China

## 1. Introduction

Freshwater accounts for only 2.5% of the earth's total water resources, whereas seawater accounts for 97.5% [1]. Seawater desalination is an important measure for alleviating the water shortage crisis. It is a realistic choice to resolve the water shortage problem by developing technologies for desalination and comprehensive utilization in coastal areas [2]. The study of seawater utilization has three key areas. One of these areas is desalination itself, which means using desalination methods to produce Freshwater [3,4]. The second area is the direct use of seawater, which refers to employing seawater instead of Freshwater as industrial cooling water, domestic water [5]. The third area is the comprehensive utilization of seawater chemical resources; namely, extracting chemical elements from seawater, and deep processing [6].

By the end of 2015, more than 150 countries and regions around the world had instituted desalination [7]. The total desalination capacity is currently approximately $8.89 \times 10^7$ m$^3$/day [8]. The top five countries in terms of desalination capacity are KSA (Kingdom of Saudi Arabia), UAE (The United Arab Emirates), USA, China, Israel [9]. In particular, in the Middle East, seawater

desalination is a vital and dependable source of Freshwater in countries such as Saudi Arabia, the United Arab Emirates, and Kuwait [10]. Furthermore, it is likely that desalination will continue to grow in popularity in the Middle East [11]. Overall, it is estimated that over $2 \times 10^8$ people worldwide obtain Freshwater by desalinating seawater or brackish water [12]. Saudi Arabia is the world's largest user of desalination and it has a capacity of $5.25 \times 10^6$ m$^3$/day, which accounts for approximately 25.9% of the world's total seawater desalination [13]. Saudi Arabia has the world's largest MSF (multi-stage flash distillation)/RO (reverse osmosis) desalination plant, the "Ras Al Khair factory" [14,15]; the world's largest MED (low-temperature multi-effect distillation) desalination plant, the "Marafiq Jubail IWPP project" [16]; the world's largest single set of MED desalination devices, the "Shoaiba II project" [17]; the world's largest MSF seawater desalination plant, the "Shoaiba III factory"; and the world's largest solar desalination plant. As the second largest user of desalination in the world, the United Arab Emirates has built more than 30 sets of desalination plants with an annual production capacity as high as $3 \times 10^8$ m$^3$ [15]. Eighty per cent of Israel's potable water is produced by desalination as of 2017 [15]. The Tathagata Centre in Fukuoka is Japan's biggest water desalination facility, and the desalinated water it produces accounts for approximately 1/12 of the country's total water supply [18]. In addition, all capital cities in Australia have desalination plants; however, they are not being regularly used due to the end of a drought. The exception is Perth where the desalination plants are regularly used [19].

The Freshwater resources per capita reflect a shortage in China [20]. The coastal areas in China are economically developed, however, these areas have large populations [20]. This leads to a low water resource amount per capita in these areas, limiting the development of social and economic conditions [21]. By the end of 2014, the 11 coastal provinces of China (including autonomous regions and directly controlled municipalities, the geographic locations of the areas could be found in Figure 1a) accounted for 30.2% of the country's Freshwater resources, fed approximately 43.4% of the population, and generated 54.7% of the GDP [22]. The recent distribution of regional water resources, population and GDP in 2017 is shown in Figure 1b–d. The contradiction between domestic, agricultural, and industrial water consumption and water in the environment is obvious. Excessive exploitation of groundwater still exists, which leads to problems, such as declines in water levels, surface subsidence, and deterioration of the ecological environment. The safety of the water supply cannot be guaranteed in an emergency due to the single available water resource. Therefore, the positive development and utilization of seawater can relieve water shortages in the coastal areas of China and can also improve the water environment and the security and reliability of the water supply. Moreover, islands can be exploited to develop the marine economy and strategy [23].

The overall scale of built desalination projects has grown continuously in China. According to the China Desalination Yearbook 2016–2017 release by the Desalination Branch of China Water Enterprises Confederation, by the end of 2017, 136 desalination projects had been completed and water production had been scaled up to $1.19 \times 10^6$ m$^3$/day [24]. At the same time, in nuclear and thermal power plants and in the petrochemical industries, seawater cooling technology had been used widely in the coastal areas [25]. The annual amount of seawater used as cooling water had reached $1.1257 \times 10^{11}$ m$^3$ [26]. The desalination technology adopted in China (a combination of imported technology and homemade technology) is becoming mature due to nearly 50 years of research, development and demonstration projects [27]. This work has laid a good foundation for large-scale applications and has placed China at the forefront of advanced seawater desalination technologies, along with the USA, France, Japan, and Israel [10,11,28]. Liu et al. [29] compared Multiple Effect Distillation (MED) and Electrodialysis method (ED) and found that MED is competitive in the production of boiler make-up water for coal-fired power plants if the cost of steam generation is ignored, and Reverse Osmosis (RO) is favoured for providing municipal water for public use. Generally, the factor that limits seawater desalination applications is its relatively high cost [30,31]. The discharge of concentrated seawater damages seashore environments [32]. Nie and Tao [33] simulated the impact of a seawater desalination system with a hydraulic and water quality model and found that the area that reflects increased salinity after a 10-day continuous discharge is four times the one that results from a 3-day continuous discharge. In addition,

the waste heat discharged by the seawater desalination process could increase seawater temperature and result in harmful algal blooms [34]. Chae et al. studied the mass reproduction of plankton caused by cooling seawater discharge, which in turn caused blockages and caused the power generation capacity of power plants to decrease [35]. There was also a study to show that the environmental issues related to seawater intake have an adverse effect on the plankton population [36]. For example, Jones and Campbell [36] showed that direct intake of surface seawater is hampered by impingement and entrainment of planktonic organisms that require additional filtration and pretreatment.

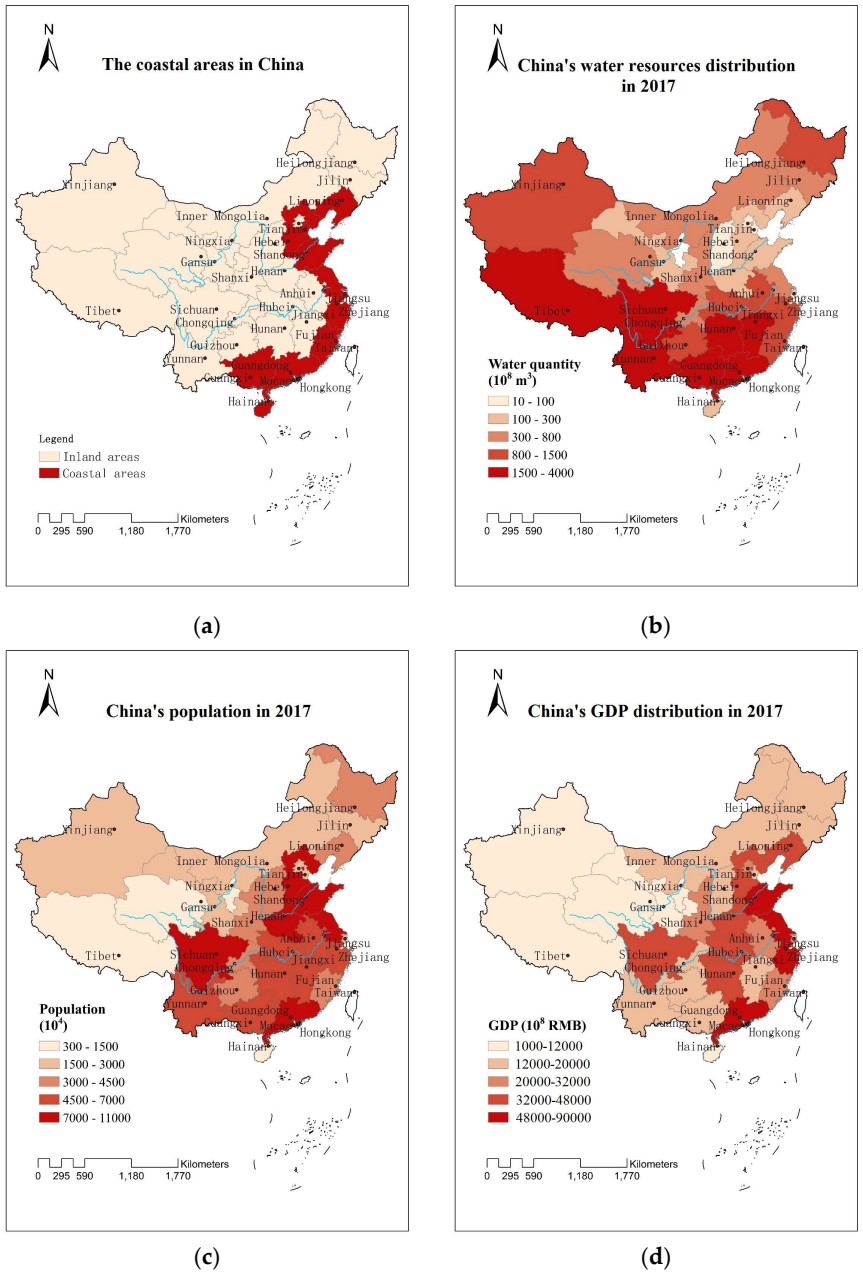

**Figure 1.** The distribution of regional water resources, population and GDP in China. (**a**) Description of the coastal areas in China; (**b**) description of China's water resource distribution in 2017; (**c**) description of China's population in 2017; (**d**) description of China's GDP distribution in 2017. The data source for Figure 1a,c,d is China Statistical Yearbook, released by the National Bureau of Statistics, China [22]. The data source for Figure 1b is China Water Resources Statistical Yearbook, released by China Ministry of Water Resources of People's Republic of China [37].

In addition, some scholars have also studied future development trends, as well as the problems and management policies associated with desalination research, in China. Zhu et al. [38] predicted that, in 2030, the total demand for desalinated seawater will be up to $1.44 \times 10^9$ m$^3$/year in the coastal areas of China. This amount corresponds to 0.52% of the coastal water demand and 0.21% of the national total water demand. The quantity of seawater utilized directly will be $1.789 \times 10^{11}$ m$^3$/year. Chen et al. [39] noted that problems still exist in the process of seawater desalination. For example, the supporting policies are poor, the independent industry is developing slowly, and the management system is inefficient. They also put forward several corresponding suggestions, including putting seawater utilization into the system of water resources allocation projects, supporting it as a public water conservancy engineering project, constructing a production-learning-research platform, setting up a special fund, and carrying out demonstration projects. Deng et al. [40] discussed three situations combining seawater desalination and energy system utilization to solve the problems of the lack of freshwater and low efficient use of low-temperature thermal energy in coastland areas.

Zhu et al. [15] summarized the status of seawater desalination in China in 2015. However, the current utilization status of desalinated seawater keeps unclear since the seawater desalination industry is developing rapidly in China and plays a more important role in relieving the water crisis especially in coastal areas of China. More importantly, the direct utilization issues of seawater and the utilization of seawater as chemical resources in China, which are processing fast and becoming more important, were not mentioned in the work of Zhu et al. [15]. Besides, a deep discussion regarding the restrictive factors, which hinder the development of the seawater desalination industry in China and a detailed presentation of some niche targeting and practical countermeasures seem also insufficient in Zhu et al. [15]. Therefore, this study introduces three aspects of the comprehensive utilization of seawater in China, including desalination, the direct use of seawater, and the use of seawater as a chemical resource. Based on this analysis of the comprehensive utilization of seawater in 2017, the study also presents the concept of optimizing the utilization of seawater from a technological perspective. Moreover, a deep exploration of the factors that restrict the development of seawater desalination and the potential countermeasures are presented in this study. This study could provide some reference for other countries facing similar water shortage state.

## 2. Seawater Desalination in China

### 2.1. The Course of Desalination Development

China attaches great importance to technology research and the development of seawater desalination. Since the "7th Five-Year Period" (1986~1990), China has supported the related technological research and development [34]. Seawater desalination was included in the "National Medium and Long-term Plan Outline of Science and Technology Development (2006–2020)" as a priority topic [41]. In 2012, the "Key Special Planning on Science and Technology Development of Seawater Desalination in the 12th Five-Year Period" was promulgated and implemented [34]. Seawater utilization has also been included in the "Recommendations for the 13th Five-Year Plan for Economic and Social Development by the Central Committee of the Chinese Communist Party (CCP)", the "Opinion of Accelerating the Construction of Ecological Civilization by the Central Committee of the CCP and the State Council" and the "Action Plan for Water Pollution Prevention and Control by the State Council" in 2015 [15]. The State Oceanic Administration has launched preliminary research on the "13th Five-Year" seawater use plan, and the pilot research focused on importing desalinated seawater into water allocation systems [15,34]. Currently, nine coastal provinces (including autonomous regions and directly controlled municipalities) have released policies relating to seawater utilization, and these policies could be found in the work of Zhu et al. [15]. These policies have led to the rapid development of seawater use technology and engineering ability and have alleviated the contradiction between the supply of water resources and the corresponding demand in coastal areas [41]. During the period of 1958–1992, studies were carried out on electrodialysis (ED)-based,

MED-based and RO-based desalination methods [34]. Since 1997, desalination development has entered the practical application stage [34]. A major scientific research project, the "RO desalination demonstration project with a capacity of 500 m$^3$/day" was completed and put into operation in Shengsi County, Zhejiang Province [15]. In 2000, with the support of a key scientific research project, the "RO desalination system and engineering technology project with a daily capacity of a thousand cubic metres" funded by the Ministry of Science and Technology of China was built on Chang Island of Shandong Province and Shengsi County of Zhejiang Province [15]. This RO demonstration project had a capacity of 1000 m$^3$/day [42]. In 2003, the first stage unit with a capacity of 5000 m$^3$/day was completed and put into operation, belonging to a high-tech industrialization project supported by the National Development and Reform Commission, the "RO desalination demonstration project with a daily capacity of ten thousand tons in Rongcheng, Shandong Province" [34]. In 2004, a 3000 m$^3$/day MED desalination device was set up in the Huangdao power plant in Qingdao. This effort was supported by the "MED desalination demonstration engineering" project proposed by the Ministry of Science and Technology of China [15]. The desalination capacity in China totalled approximately 140,000 m$^3$ per day in 1980–2006 [24].

The major technologies used for seawater desalination in China are the RO, MED and MSF methods [34]. The main desalination methods and development trends can be seen in Table 1. The RO projects account for 86% of the total desalination projects, and their capacity accounts for 63% of the total desalination capacity in China [15,34]. Furthermore, new technologies, such as solar distillation, have been used in small- and medium-scale practical applications in the experimental stages [43]. The first solar desalination demonstration project in China has been constructed in Hainan Province [43].

**Table 1.** Main desalination methods and development trends in China.

| Methods | Status Quo | Development Trend |
|---|---|---|
| Reverse osmosis desalination [44] | Practical application | Semipermeable membrane and membrane module |
| | | New technology and energy recovery |
| Low-temperature multi-effect distillation [44] | Practical application | Thermodynamics and fluid mechanics |
| | | Boiler-scale control |
| | | Materials and equipment |
| | | Combined with multi-stage flash distillation |
| Multi-stage flash distillation [44] | Practical application | Combined with nuclear power generation to set up large scale devices |
| Vapour compression distillation [45] | Practical application | Small and medium size |
| Solar distillation [45] | Research and development, small test and application | Applied in areas with strong sunlight |
| Crystallization method [45] | Research and development, small and medium test factory | Mud generation, transmission and thin ice separation and washing |
| | | Choice and recycling of solvent and water mixture |
| Electrodialysis method [45] | Practical application in desalination and marine salt producing | Membrane |
| | | High-temperature electrodialysis method |
| | | Combining desalination with comprehensive utilization |
| Solvent extraction method [45] | Research and development | Choice and recycling of solvent |
| Ion exchange method [45] | Practical application in pure water production | Resin synthesis and regeneration |

### 2.2. The Status Quo of Desalination Development

The increase in capacity of Chinese desalination projects from 1980 to 2017 is shown in Figure 2. Since 2006, a series of desalination projects have been constructed with capacities of ten thousand cubic metres/day and above. The scale of single projects has been continuously increasing. The capacity has reached a stage with rapid development since 2009 [34]. Moreover, the building of new capacity has shown a growth trend from 2012 to 2013 due to the construction of the Baifa desalination project of Qingdao, which has a capacity of 100,000 m$^3$/day, and the Beijiang first phase project, which has a

capacity of 100,000 m$^3$/day, in Tianjin [15,34]. These projects have caused the output to increase during these two years. As shown by Figure 2, by the end of 2017, the capacity of desalination plants in China had reached 1.19 × 10$^6$ m$^3$/day [20]. Comparing 2015 and 2017, it could be seen that a large increase of 180,000 m$^3$/day in desalination capacity has been made recently, showing a fast development speed of seawater desalination industry in China [24].

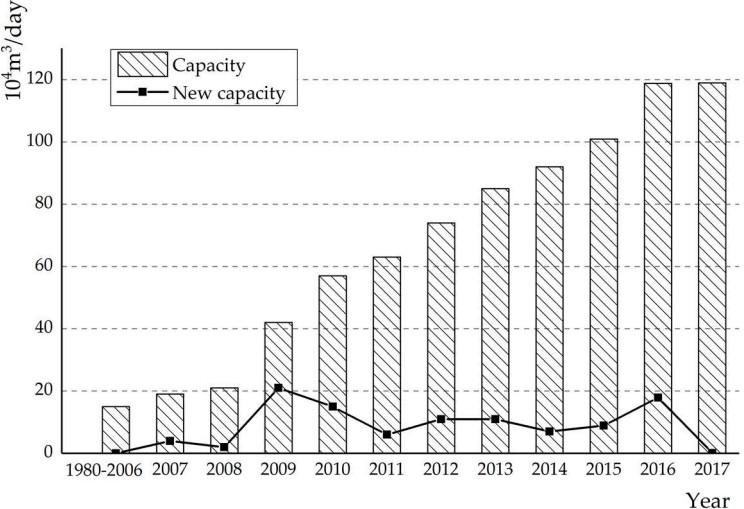

**Figure 2.** Changes in the capacity of Chinese desalination projects from 1980 to 2017.Data Source: Desalination Branch of China Water Enterprises Confederation, "China Desalination Yearbook 2016–2017" [24]. The data by the end of 2015 was also presented in Zhu et al. [15].

The construction number of desalination projects in China from 1980 to 2017 can be seen in Figure 3. Similarly, in 2006, the number of projects reflected rapid development. The number of completed projects exceeded one hundred for the first time in 2013. As this figure shows, by the end of 2017, China had been built 136 desalination engineering projects [24] The largest desalination projects have capacities of 200,000 m$^3$/day [24]. Comparing 2015 and 2017, it could be observed that about 15 seawater desalination plants have been constructed recently, indicating that the seawater desalination industry of China is in a rapidly developing state [15,24].

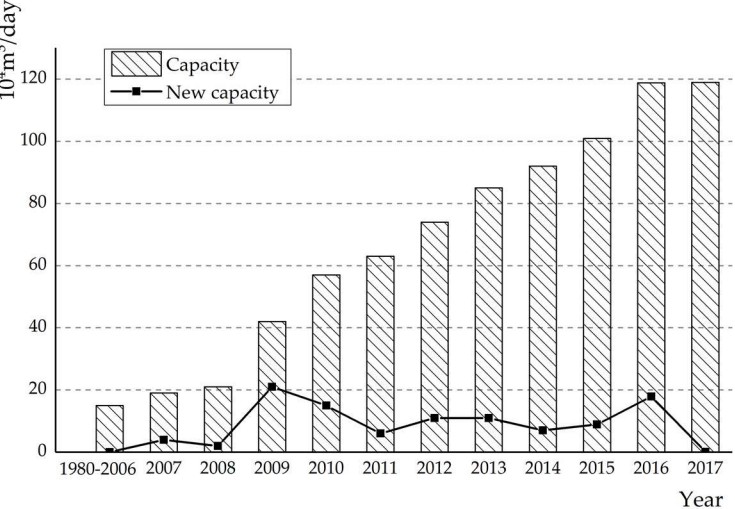

**Figure 3.** The construction of Chinese desalination projects from 1980 to 2017. Data Source: Desalination Branch of China Water Enterprises Confederation, "China Desalination Yearbook 2016–2017" [24]. The data by the end of 2015 was also presented in Zhu et al. [15].

The spatial distribution of the Chinese regional desalination capacity and the number of projects in 2017 is presented in Figure 4. The construction of seawater desalination plants was mainly concentrated in Zhejiang, Shandong, Tianjin, Hebei and Liaoning. The capacity construction of desalination projects in Tianjin and Shandong Provinces were prominent, accounting for 26.68% and 23.77% of the national projects, respectively [24]. As the primary builder of plants, Zhejiang has built 41 desalination projects. The number of desalination projects in Tianjin was not large; however, the capacity of these projects was remarkable [24]. In addition, large-scale desalination work has not been carried out in the Guangxi autonomous region or in Shanghai. This lack of development is attributed to the abundant water resources in those areas and other factors [34].

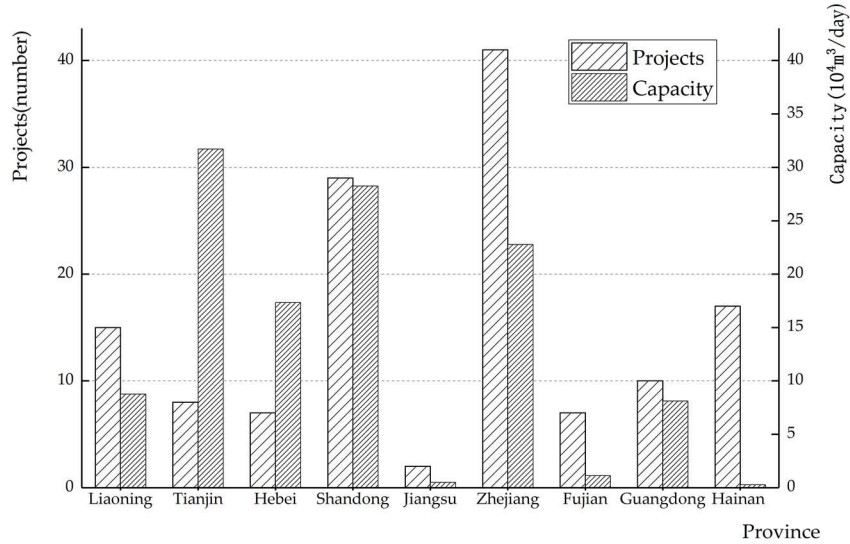

**Figure 4.** The spatial distribution of seawater desalination capacity and projects in China in 2017. Data Source: Desalination Branch of China Water Enterprises Confederation, "China Desalination Yearbook 2016–2017" [24].

The desalination project with the largest capacity in China, the Beijiang power plant in Tianjin, uses MED desalination technology, which has advantages such as high efficiency, low cost, good anticorrosion performance, and strong adaptability [24]. Its daily capacity can reach 200,000 m$^3$, including 20,000 m$^3$ of capacity for self-use, and the remaining 180,000 m$^3$ is allocated to Tanggu (90,000 m$^3$), Hangu (20,000 m$^3$), and the developing zones (70,000 m$^3$) [24]. The desalinated seawater flows into the water plant, supplying the city's domestic water supply through the municipal pipe network. The Baifa desalination plant in Qingdao is the largest in China [15,24]. Its daily capacity of 100,000 m$^3$ has been placed into practical use. The project incorporates the world's advanced desalination technology, which is Ultrafilters RO desalination technology [46]. Its desalinated water is used for drinking and has accounted for 15–20% of the urban water supply of Qingdao, most of which has been used in industrial and commercial enterprises. The Dagang Xinquan desalination plant was built in 2009 in Tianjin and uses RO desalination technology [15,24]. Its desalinated water has mainly been used to supply industrial projects in Dagang, especially the production of ethylene, which receives one million cubic metres of water, relieving the tense situation associated with the regional water supply [15].

Figure 5 shows a recent distribution of the utilization of desalinated seawater in 2017. As this figure shows, desalinated water is primarily used in the domestic water supply (33.11%), the power electric generation (31.585), and the manufacturing of petrochemicals (12.29%). Iron and steel, chemical industry and nuclear power account for 13.03%, 5.05% and 4.61% respectively [24]. Other objectives, including shipping, harbour services, testing, buildings and so on, only accounts for 0.33%. This is slightly different from the findings of Zhu et al. [15].

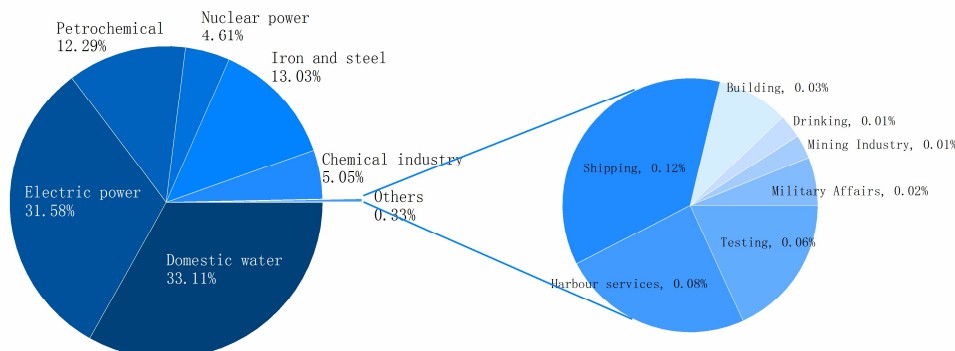

**Figure 5.** The distribution structure of utilization type of desalinated water in 2017. Data Source: Desalination Branch of China Water Enterprises Confederation, "China Desalination Yearbook 2016–2017" [24].

Some studies showed that distillation and reverse osmosis are the most widely used techniques worldwide [47–52]. Thermal processes for seawater distillation have been developed in countries where the thermal plant can be linked to a power plant, which acts as a low-temperature heating source [30]. The major thermal processes are MSF and MED [53–56]. As the international mainstream technologies of seawater desalination, the RO, MED and MSF seawater desalination technologies have been applied commercially. Figure 6 shows the plants and capacity of desalination using various technologies by the end of 2017 [24]. It is evident that RO still maintains its mainstream status. By the end of 2017, 117 plants had applied RO technology to desalinize seawater, whereas 16 plants had applied MED technology. The capacity of desalination plants that use RO technology is 813,600 $m^3$/day, whereas the MED technology is used in desalination plants with a total capacity of 369,100 $m^3$/day. Moreover, the numbers of desalination plants that use the MSF and ED technology are one and two respectively [24]. The capacity of the desalination plant that uses MSF technology is 6000 $m^3$/day, whereas that of the ED-based plant is 300 $m^3$/day [20]. Comparing the data in 2015 [15] and 2017, we could find that the capacity using RO increases 153,600 $m^3$/day, which is a large progress. Whereas the capacity using MED only increases 9100 $m^3$/day. These show the potential of RO technology in the seawater desalination industry of China. It should also point out that the forward osmosis and capacitive osmosis is advancing rapidly [57,58]. Rapid development has been made in Chinese agriculture in terms of increasing salt tolerance of major food crops, the use of diluted saline water for irrigation and so on [59].

There are also some works on the economic effect of seawater desalination [60,61]. For example, Zou and Liu [61] carried out an analysis regarding the economic effect of different methods. The sea ice desalination technology and the desalination of oil-field wastewater via vacuum membrane distillation (VMD) have been also made a progress in China [62,63].

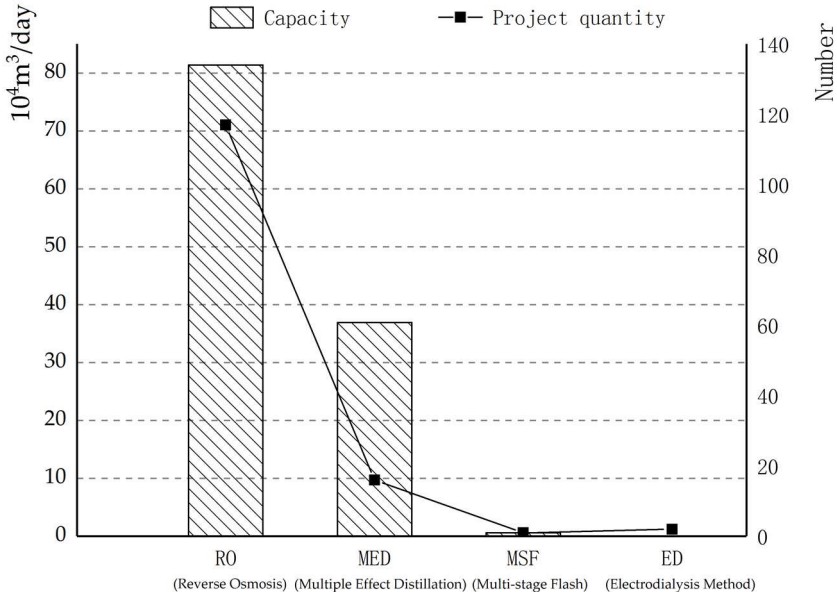

**Figure 6.** The distribution structure of utilization type of desalination technology by the end of 2017. Data Source: Desalination Branch of China Water Enterprises Confederation, "China Desalination Yearbook 2016–2017" [24]. The data by the end of 2015 was also presented in Zhu et al. [15].

## 3. Direct Utilization of Seawater in China

Direct utilization of seawater is the technology and process of using seawater as industrial cooling and flushing water to replace the use of scarce Freshwater [25]. It also includes seawater desulfurization, seawater irrigation, and the use of seawater in heat pumps.

### 3.1. The Development Status Quo of Once-Through Seawater Cooling

In China, the technology for the direct utilization of seawater mainly includes once-through seawater cooling and circulating seawater cooling and is dominated by once-through seawater cooling [25]. In coastal areas, it is widely used in nuclear and thermal power generation and the petrochemical industry, and there is an upward trend in the scale of plants [25]. The application of once-through seawater cooling in China has approximately 70 years of history. Figure 7 shows the annual amount of direct utilization of seawater in China by the end of 2017 according to the data "the National Seawater Utilization Report in 2017" released by State Oceanic Administration of China [25]. From it, it could be seen that from the beginning of 2006, the technology of once-through seawater cooling entered a stage of rapid development. For the first time, the annual utilization was greater than $10^{11}$ m$^3$ in 2014 [25]. By the end of 2017, the amount of direct utilization of seawater reached $1.347 \times 10^{11}$ m$^3$.

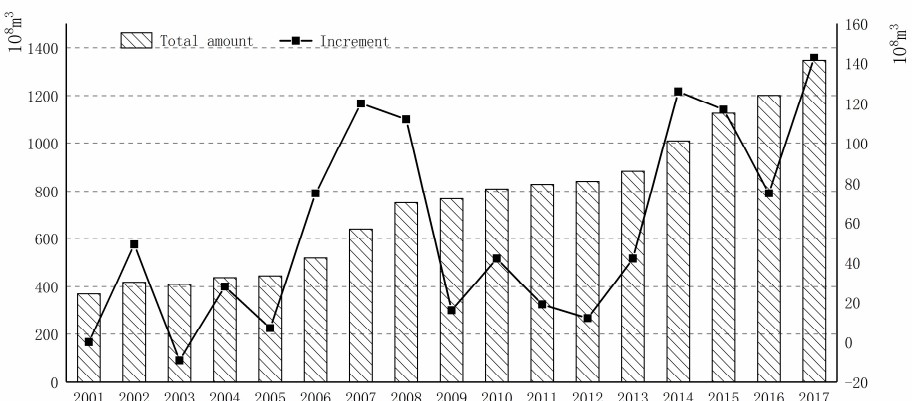

**Figure 7.** The annual amount and increase in seawater used directly in cooling projects in China. Data Source: State Oceanic Administration of China "The National Seawater Utilization Report in 2017".

The 11 coastal provincial administrative regions of China all have seawater cooling projects, including Guangdong, Zhenjiang, Fujian, and Liaoning. Figure 8 shows the utilization amount and proportion of seawater cooling in 2017 according to the data "the National Seawater Utilization Report in 2017" released by State Oceanic Administration of China [25]. In 2017, the annual seawater utilization in Guangdong, Zhenjiang, Fujian, Liaoning Provinces were $4.18 \times 10^{10}$ m$^3$, $3.07 \times 10^{10}$ m$^3$, $2.26 \times 10^{10}$ m$^3$ and $0.92 \times 10^{10}$ m$^3$, respectively.

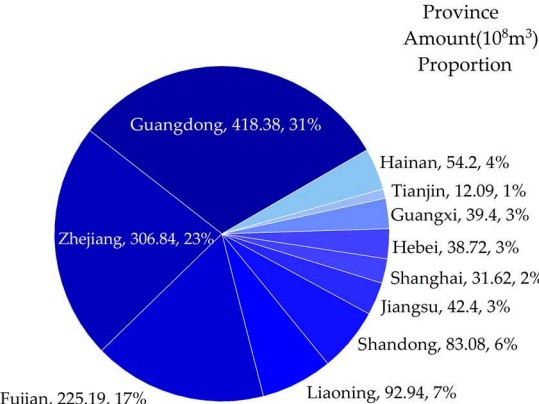

**Figure 8.** The amount and proportion of seawater used directly in cooling projects in the 11 coastal regions of China in 2017. Data Source: State Oceanic Administration, People's Republic of China "The National Seawater Utilization Report in 2017" [25].

*3.2. Direct Utilization of Seawater in Other Ways*

Domestic seawater technology belongs to the category of direct seawater utilization. The main principle involves using seawater to replace Freshwater directly as domestic water (which is mainly used for flushing). As an effective method to alleviate the tension of Freshwater shortages, domestic seawater technology is significant in the conservation of water in coastal cities. By 2012, approximately 80% of the population of Hong Kong used seawater and the annual seawater supply had reached $2.72 \times 10^8$ m$^3$/year [64]. Due to the large amount of water supplied and the reduced cost, seawater is free of charge to users in Hong Kong. After addressing the key problems through production, learning, and research over many years, China has made progress in the construction of demonstration projects that show the potential of domestic seawater technology. In 2007, a demonstration project was built in the Qingdao Longhai residential community with a total construction area of 460,000 m$^2$. All of its technical indexes achieve or exceed the performance targets. Thus, this project establishes a solid

foundation for the promotion of domestic seawater technology in China. There is also a study to mention the treatment (biological contact oxidation) of domestic seawater [65].

Seawater desulfurization has several advantages, such as simplicity, reliability, a high desulfurization efficiency and low operating costs. The first set of seawater desulfurization devices in China was built in the Shenzhen Xibu power plant in 1997. At present, there are nearly 50 sets of devices running, and they are widely used in industrial furnace and power plant flue gas desulfurization [66]. Seawater desulfurization technology is suitable for the comprehensive technical level and operation management level in China, so its application potential is tremendous [67]. China has carried out cultivation experiments using Salicornia, rice straw and other salt-tolerant plants, as well as salt tolerance experiments using Vigna unguiculata, tomatoes, rice and other economic crops and grains, thus achieving considerable economic and ecological benefit [2]. In addition, seawater can be used as a source for heat pumps. Seawater at a certain depth is extracted as a heat source for heating in winter or a cold source for air conditioning and refrigeration in summer. The Qingdao power plant built the first Chinese seawater source heat pump in 2004 in order to save energy [68].

Environmental circulating seawater cooling technology has been developed based on circulating fresh-water cooling and once-through seawater cooling technology. The development of circulating seawater cooling technology started relatively late in China; however, the speed of development has been high. The related research and application were carried out in the "8th Five-Year Period". A medium-term pilot study was completed in the "9th Five-Year Period", which involved circulating seawater cooling at a scale of 100 $m^3$. Thanks to the national science and technology support plan since the "10th Five-Year Period" and the "11th Five-Year Period", seawater circulating cooling technology has achieved a major breakthrough. The projects with circulating seawater cooling technology are mainly distributed in Tianjin, Hebei, Shandong, Zhejiang and Guangdong. The first demonstration project with a scale of 1000 $m^3$ in the chemical industry was completed in the Tianjin soda factory in 2004. Furthermore, the first demonstration project with a scale of 10,000 $m^3$ in the electric power industry was built in the Zhejiang Ninghai power plant in 2009. Through technological research, practical problems were effectively solved in circulating seawater cooling systems involving seawater corrosion, scaling, biological adhesion, salt deposition and salt fog spatter in seawater cooling towers. The technology has achieved many innovations, such as using seawater to replace Freshwater in industrial cooling and reducing pollution by more than 95%, compared to once-through seawater cooling technologies by using ordinary carbon steel in the cooling system. The relevant technologies have provided several missing pieces [65].

## 4. Utilization of Seawater as a Chemical Resource and Structural Optimization in China

### 4.1. Utilization of Seawater as a Chemical Resource

Retentate brine is the concentrate that remains after separating the Freshwater in the process of seawater desalination. Retentate brine is characterized by high salt content, which is approximately 500–12,000 mg/L [69,70], and mainly includes $Na^+$, $Ca^{2+}$, $Mg^{2+}$, $Cl^-$, and $SO_4^{2-}$. China's sea salt extraction originated in the Song Dynasty, dating back more than a thousand years [71]. If a large amount of untreated retentate brine is directly discharged into municipal pipelines or dumped into the sea, it will not only cause adverse effects on the seawater environment but also waste the mineral content of the brine.

As an innovative economic development mode that developed against the background of the increasingly serious contradictions between economic development, resource shortages and energy and environmental pollution after human society entered the middle and later periods of industrialization, the idea of sustainable development appeared. "Circular economy" is a generic term that describes the quantitative reduction, reutilization and resource reuse activities in areas such as production, circulation and consumption. With the effective and cyclic utilization of resources, it is basically characterized by high efficiency, low consumption and low pollution. Its essence is the pursuit of

more effective resource utilization, reduced waste discharge and higher economic benefit. Through the development of the circular economy mode, abandoned materials are reused in the process of production until no available resources are contained in the retentate brine to achieve "minimum quantitative" or zero discharge [72]. Zero discharge of strong brine means concentrating the strong brine again, increasing the salinity to 50,000–80,000 mg/L [73], improving the Freshwater recovery rate to 80% or more, and decreasing the volume by 4/5, as well as recycling part of the Freshwater. Last, a small amount of the concentrated remaining brine can also be used comprehensively, for example, refining and crystallization treatment of high-concentration brine can achieve zero discharge of industrial concentrated brine [74,75].

The Chinese government attaches great importance to new technological research and development involving the comprehensive utilization of seawater. In the "National Medium and Long-term Plan Outline of Science and Technology Development (2006–2020)", "technology for the direct use of seawater" and "technology for the use of seawater as a chemical resource" are listed as priority topics in the field of "water and mineral resources". These two technologies were listed in the "9th Five-Year", "10th Five-Year" and "11th Five-Year" science and technology support plans (which address key problems) developed by the National Ministry of Science and Technology, and thus have received key support. The utilization of seawater as a chemical resource mainly involves extracting salt, potassium, bromine and magnesium, as well as other materials. The main production enterprises are located in Shandong, Jiangsu, Hebei, Tianjin, and Liaoning.

As shown by Figure 9 according to the Statistical Yearbook of China (2009–2017) released by the National Bureau of Statistics of China, by the end of 2017, crude salt production in China had reached $3.33 \times 10^7$ ton [22]. The annual production in Shandong Province was the greatest among the 11 coastal areas (see Figure 9), accounting for about half of the total national production, mainly due to its well-developed seawater desalination industry [22]. Moreover, more than 80% of the crude salt produced is consumed by the NaOH and $Na_2CO_3$ industries. As presented in Table 2, the annual production of NaOH and $Na_2CO_3$ is developing rapidly. By the end of 2017, NaOH production reached $3.33 \times 10^7$ ton, and $Na_2CO_3$ production exceeded $2.77 \times 10^7$ ton respectively [22]. These uses are considered to represent an important part of the comprehensive utilization of seawater in the recycling industry chain.

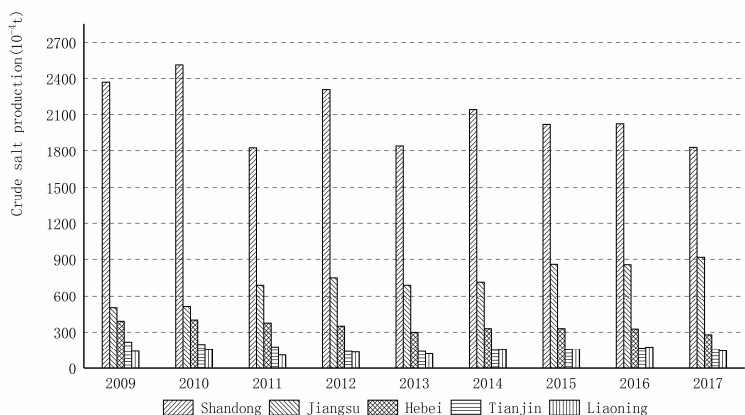

**Figure 9.** The crude salt production in the five largest areas during 2009–2017 in China. Data Source: National Bureau of Statistics of the People's Republic of China, "Statistical Yearbook of China, 2009~2017" [22].

**Table 2.** The production ($10^4$ t) of NaOH and $Na_2CO_3$ during 2009–2017 in China [1].

|  | 2009 | 2010 | 2011 | 2012 | 2013 | 2014 | 2015 | 2016 | 2017 |
|---|---|---|---|---|---|---|---|---|---|
| **NaOH** | 1832.37 | 2228.39 | 2473.52 | 2696.82 | 2927.44 | 3063.51 | 3020.66 | 3201.68 | 3329.17 |
| **Na$_2$CO$_3$** | 1944.77 | 2034.82 | 2294.03 | 2395.93 | 2431.63 | 2525.84 | 2591.80 | 2584.98 | 2767.14 |

[1] Data Source: National Bureau of Statistics of China, "Statistical Yearbook of China, 2009–2017" [22].

### 4.2. The Optimization of the Recycling Industry Chain

As a technology for the comprehensive utilization of seawater chemical resources, the traditional halogen method can produce salt, magnesium sulfate, potassium chloride, bromide, and magnesium chloride. However, it has some deficiencies, namely, a long process flow, an unreasonable utilization of heat energy, considerable steam consumption, high production costs, and inadequacy for large-scale seawater desalination. The current research on zero discharge of concentrated brine is focused on membrane distillation, forward osmosis, freezing, and spray drying methods, as well as other techniques [75]. Combined with the low cost and energy consumption technologies mentioned above, this paper puts forward a process for the comprehensive utilization of retentate brine that optimizes the original industrial structure. The steps are as follows:

1.  Taking seawater as a raw material, obtain Freshwater and retentate brine through the desalination process.
2.  Produce bromine by applying the air blow-out method to retentate brine and obtain debrominated brine.
3.  Add NaOH and $Na_2CO_3$ to the debrominated brine to produce magnesium hydroxide and calcium carbonate and refine the brine.
4.  Produce sodium sulfate decahydrate by applying the freezing method to the refined brine and obtaining desulfurized mother liquor.
5.  Produce sodium chloride by applying high-temperature evaporation to the desulfurized mother liquor and obtaining the mother liquor of salt.
6.  Produce potassium chloride and potassium mother liquor by low-temperature evaporation. The mother liquor is then circulated into the extraction process of sodium sulfate by the freezing method.

As Figure 10 shows, the industrial chain (see Figure 10) is characterized by reasonable technological processes and cold utilization, improved value-added products, low production costs and low energy consumption [76]. Thus, it is adapted for large-scale seawater desalination.

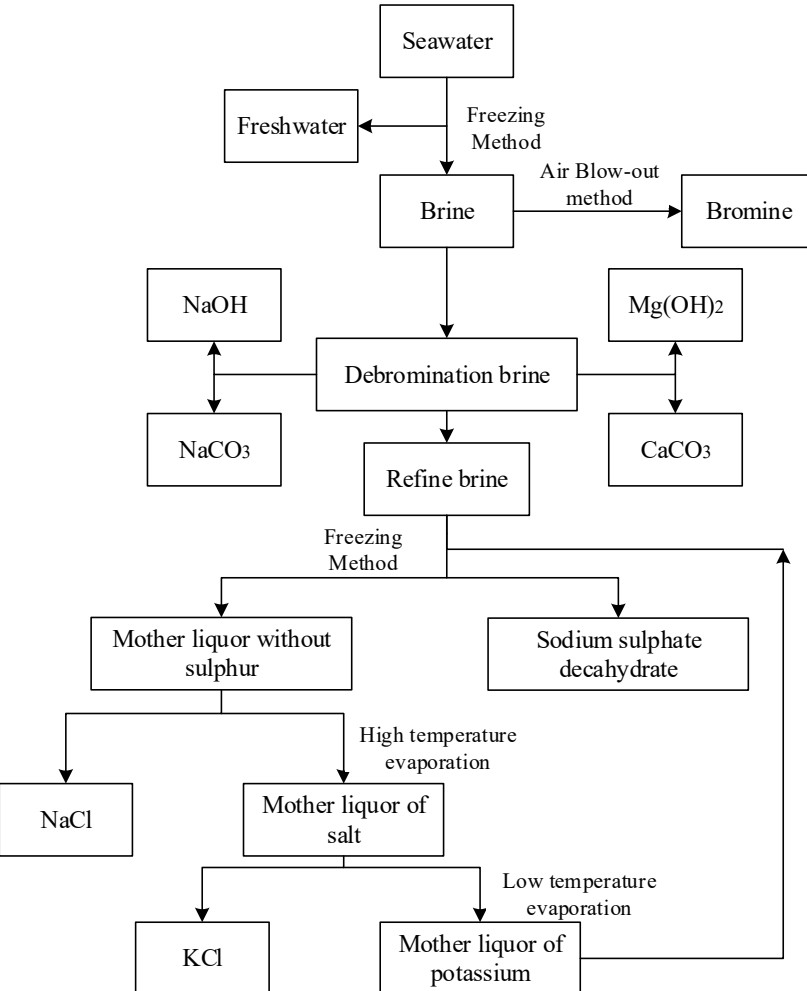

**Figure 10.** The optimized industry chain of seawater chemical resource recycling utilization [77].

As it employs the circular economy model of "making power generation play the leading role and integrating power generation, heat supply, seawater desalination, salt production through concentrating seawater, the comprehensive utilization of bittern and wastes, and the arrangement of the land economy" [78], the Tianjin Beijiang power plant project is known as the "Beijiang model", and its combination of electric power, water and salt is the first example of its kind in China. After the first phase of the project was put into production, the Binhai New Area, the Bohai economic circle, Beijing, Tianjin and Tangshan could be provided with $1.1 \times 10^{10}$ kilowatt-hours of electricity every year by the Beijiang power plant. By using the waste heat from power generation for seawater desalination, seawater desalination devices were constructed with a daily capacity of 200,000 m$^3$. The concentrated seawater after desalination is introduced into the Tianjin Hangu saltern, greatly enhancing the efficiency of salt production, and the annual output can be improved to approximately 500,000 m$^3$. The mother liquor is employed in the process of chemical production, and the annual output of salt is 150,000 m$^3$. In addition, the comprehensive utilization process also produces 2600 m$^3$ of bromide, 3500 m$^3$ of potassium chloride, 3.5 m$^3$ of magnesium chloride and 5000 m$^3$ of magnesium sulfate. Through the use of concentrated seawater to make salt, the 22 square kilometres of reserved land used as salterns will be changed into land for construction. This land will thus see a great increase in the value of its use and promises to be a huge development space for the construction of the Binhai New Area. In addition, the comprehensive waste utilization programme is predicted to generate approximately 750,000 m$^3$ of fly ash, 150,000 m$^3$ of slag and approximately 200,000 m$^3$ of FGD (flue

gas desulphurisation) gypsum every year. Through the production of building materials from the waste, comprehensive utilization can be realized [79].

The production of sea salt in China is approximately $3 \times 10^7$ m$^3$. The production capacity of bromide extraction is 150,000 m$^3$ using concentrated seawater. China has built the first international potassium extraction project that uses seawater with a scale of 10,000 m$^3$, leading to the achievement of technical and economic breakthroughs [64]. New technologies have made periodic progress, including magnesium extraction, marine inorganic functional materials and the comprehensive utilization of concentrated seawater. The technology and standard system for the comprehensive utilization of seawater have been initially developed in China, promoting the development and support of emerging industries that employ seawater resources.

## 5. Restrictive Factors and Potential Countermeasures for the Comprehensive Utilization of Desalinated Seawater in China

Zhu et al. [15] mentioned that there are some problems related to the seawater desalination in China, the including high cost of desalinated seawater, low industrialization of desalination technology, and environmental issues. However, a deep and systematic analysis regarding those factors that restrict the further development of China's seawater desalination, and especially the detailed presentation of some niche targeting and practical countermeasures, which could be referred to by some administrators, was not presented in the work of Zhu et al. [15]. Here, we attempt to summarize five restrictive factors as presented below, including low localization of key technologies, high economic cost, insufficient water production rate, environmental problems, and lack of supporting policies, and analyse the role that these factors play in hindering the seawater desalination development. For each factor, we put forward some applicable countermeasures, which could be referred to by other countries facing similar seawater desalination situations.

### 5.1. Low Localization Rate of Key Technologies and Equipment

As Zheng et al. [30] and Zhu et al. [15] have mentioned, the availability of key desalination technologies and equipment are the main factors that affect the cost of water desalination. At present, desalination projects in China that have a capacity of more than 10,000 m$^3$ mainly rely on foreign technology. For example, the Tianjin Dagang Xinquan water desalination plants all use SWC5 membrane elements produced by the Hyde Company in the US. The key materials used in desalination membranes mainly include ultrafiltration membranes, which are used in the pretreatment process, and desalination reverse osmosis membranes, which are used after pretreatment. In the case of reverse osmosis membranes, the RO method accounts for 67% of the total desalination projects. However, its core materials and key equipment, such as seawater membrane modules, energy recovery devices, high-pressure pumps and some chemical raw materials are mainly imported. According to the project equipment investment cost ratio, the localization rate is less than 50% [38]. It can be seen from the "Chinese Desalination Development, Research and Analysis Report" published by the State Technology Centre of Environment Protection and Membrane Separation Engineering that the ultrafiltration membranes used domestically are all imported. In addition, imported reverse osmosis membranes occupy 90% of the market share, and these membranes are mainly produced by The Dow Chemical Company in the United States and Asahi Kasei Corporation and Toray Corporation Industries in Japan. The manufacturers of high-pressure pumps and energy recovery devices mainly include KSB (This is the company that manufactures pumps and valves and is one of the three largest pump and valve manufacturing companies in the world.) in Germany [80], Calder in Switzerland [81], ERI (Energy Recovery, Inc.) in the United States, Torishima in Japan, and so on.

The countermeasures consisting of the low localization rate of key technologies and equipment are as follows:

1.  The "Special Plan for Seawater Utilization" introduced in 2005 proposed that the localization rate of desalination equipment should be 90% or more by 2020. Considering the current development

situation, we should carry out independent innovation in terms of process optimization, cost reduction and research into the key equipment and technologies. The leading domestic manufacturers are Hangzhou Big Dipper Membrane Products Co., Ltd., Beijing Shidai Walton Technology Co., Ltd., Bluestar Co., Ltd., Tianjin Motiammo Co., Ltd., and Hainan Lisheng Water Purification Technology Co., Ltd. The domestic market share of desalination membranes is approximately 10% [82]. The Vontron reverse osmosis membrane produced by Beijing Shidai Walton Technology Co., Ltd. has passed the NSF certification in the United States and has distributors, agents and fixed customers around the world [83,84]. Construction of the largest domestic reverse osmosis membrane project began in Beijing in August 2009, and this project is jointly built and financed by Bluestar Co., Ltd. and the Toray Company in Japan [85].

2. At the same time, seawater desalination demonstration projects should be constructed. By encouraging enterprise and scientific research institutes to incorporate the latest research achievements (especially innovations in key pieces of equipment, such as reverse osmosis membranes, energy recovery devices, high-pressure pumps, and evaporators) into demonstration projects to be tested and improved, we could gradually realize the localization of desalination equipment manufacturing. The Southern Pump Industry Company undertook the "Research, Development, Application and Demonstration of Energy Recovery Devices for Desalination" project of the "Key Equipment for Reverse Osmosis Desalination Development" national science and technology support plan during the "12th Five-Year Period", which has resulted in the local production of energy recovery devices. The research work of this company is progressing, and preparations are being made for third-party testing. In addition, the biggest single-plant high-pressure pump in China, the pump used in the Zhoushan Liuheng desalination project, which was developed by the Southern Pump Industry Company and has a daily capacity of 12,500 m$^3$ and was put into operation during the second phase of the project in 2014. Its acceptance indicates that the localization rate of the key technology of desalination membranes is further improving.

*5.2. High Economic Costs*

The main factors influencing costs consist of the equipment localization rate, energy consumption, the efficiency of laying pipelines, pretreatment, and other factors [15,34]. (1) The energy consumption accounts for more than 60% of the cost of desalinizing one cubic metre of seawater. For projects with a capacity of more than 10,000 m$^3$/day, if the heat energy consumed by the distillation method is converted into electric energy, the MSF, MED, and RO methods will consume amounts of energy equal to approximately 8.5 kw·h/m$^3$, 5.5 kw·h/m$^3$ and 4.3 kw·h/m$^3$, respectively. (2) The cost of laying the pipeline is larger for long-distance transmission. Taking the Caofeidian desalination project, which supplies water to Beijing as an example, the linear distance from Caofeidian to the fourth ring road east of Beijing is approximately 220 km, and the height difference is 45 m. The cost of laying the pipeline is 1 RMB/m$^3$, which accounts for 15% of the total cost after water transmission. (3) Whether the thermal process or the membrane method is used in seawater desalination, pretreatment of the seawater is indispensable. The pretreatment cost is one of the important factors that influence the desalination cost.

The countermeasures to the high economic cost restrictive factor are as follows:

1. To reduce the cost of energy consumption and pretreatment, the combination of nuclear power and desalination has become an important development trend. The combination of these systems meets the need for cooling water in nuclear electricity generation. On the other hand, it provides cheap energy for desalination, creating the necessary conditions for the large-scale utilization of seawater. At present, the first combined nuclear power plant and seawater desalination system built in China, the Liaoning Hongyanhe nuclear power plant, is working well. The first nuclear power project approved during the "11th Five-Year Period", the Hongyanhe nuclear power plant, has four machine sets and a scale of a million kilowatt-hours. It greatly alleviates water

shortage conditions and breaks new ground to effectively solve the problem of freshwater supply to nuclear power plants.

2. As the first city to introduce the use of desalinated seawater, Qingdao has leading significance for the nationwide implementation of this practice in terms of demonstrating the feasibility, method of operation, and the means of subsidizing and financing the project. In another case, the Caofeidian desalination project was adopted by the Hebei Development and Reform Commission in 2014. It is expected to produce water for Beijing beginning in 2019. This large-scale project, which has a daily capacity of 1,000,000 $m^3$, will meet one-third of the water demand of Beijing. The cost of the desalinated seawater will be 7–8 RMB/$m^3$, however, the cost of domestic water in Beijing is 4 RMB/$m^3$. Clearly, there is a certain gap between these water prices, which is supposed to be adjusted gradually in the future.

*5.3. Insufficient Water Production Rate of Desalination Plants*

In contrast to the rapid growth in production ability, the market demand for desalination has seen relatively slow growth. The plants are being operated below capacity, and equipment is idle to a severe degree [34]. Even in Tianjin, where the earliest large-scale seawater desalination development projects were carried out with great expectations, desalinated water has not been used to supply the municipal pipe network on a large scale. According to statistical data, in 2013, the amount of desalinated seawater used was $3.1 \times 10^7$ $m^3$ in Tianjin, which accounts for nearly 3% of the total water supply. Only a small part of the desalinated seawater is being supplied to the domestic water pipe network. The six completed and operating desalination plants all have an unused capacity problem to varying degrees. The Tianjin Beijiang power plant desalination project (phase I) has been completed, and it has a capacity of 200,000 $m^3$; however, the actual production is less than 70,000 $m^3$ per day, and its capacity utilization rate is less than 35%. Statistical data from the International Desalination Association (IDA) show that the scale of desalination projects globally had reached $8.655 \times 10^7$ $m^3$ as of September 2014, which contrasts with the idle status of the domestic projects. The average annual growth rate in desalinated seawater globally is over 8% in nearly 10 years. In addition, 60% of the total amount is directed to municipal water supplies, solving the problems related to the water needs of more than $2 \times 10^8$ people [86].

The fundamental problem of idle capacity lies in the high cost of desalination, which is related to three factors. One of these factors is that energy prices in China are high. For the power consumption associated with the same capacity, the energy cost will be high. Second, the utilization rate of desalination equipment is low. The investment depreciation and equipment maintenance cost are high per unit cubic metre of water. Third, the temperature of seawater is low in areas that lie farther north, and the pollution is serious; thus, treatment costs increase accordingly [87].

To alleviate the problem of the low water production rate, the following countermeasures are made:

1. Given the current desalination cost level, direct subsidies from the government are indispensable in solving the problems related to the large-scale idling of capacity. The attitude of the government largely determines whether seawater desalination can be carried out on large scales in urban water supply systems. As a city manager would do, the government should not just regard the water supply as a financial and technical problem. It is important to establish a safe and stable water supply security system in order to meet the demands of economic development and residents. The marginal benefits of urban economic development can fully or partially provide an incentive to supply government subsidies for the enforcement of desalination. In addition, the large-scale application of seawater desalination and equipment localization can also become a new industry, driving local economic prosperity and tax increases.

2. The coastal cities of China should implement reforms in tap water pricing, specifically categorized water pricing and ladder water pricing [88]. These reforms increase the water resource tax for water users whose consumption is excessive and indirectly increases the demand for

unconventional water resources. By calculating the water demand at different water prices, the demand for desalinated seawater can be determined at the comprehensive cost level [89]. Roseta Palma and Monteiro [90] systematically proved that, in the case of insufficient resources and different users, ladder pricing can promote resource saving. In Tianjin, the ladder pricing of domestic water, shown in Table 3, has been carried out since 1 November 2015. Incremental, ladder-based water pricing policies have the following advantages. First, they promote social justice. Monteiro [91] noted that ladder water pricing ensures the basic demands of low-income people by making the first level water price lower than the marginal cost of supplying the water. However, higher-income families who consume more water are charged extra via the higher levels of the ladder water pricing. Second, Boland and made [92] indicated that ladder water pricing tends to accelerate the effective utilization of water resources and the realization of water savings. Third, Porter [93] and Opitz et al. [94] concluded that ladder water pricing can increase economic efficiency, achieving the most effective use of resources with balanced profits and losses in the water industry.

**Table 3.** Tianjin domestic water ladder price [1].

| Level | Water Consumption ($m^3$/household·year) | Price (RMB/$m^3$) | Base Price (RMB/$m^3$) | Water Resource Fee (RMB/$m^3$) | Sewage Treatment Fee (RMB/$m^3$) |
|---|---|---|---|---|---|
| First step | 0–180 | 4.9 | 2.61 | 1.39 | 0.9 |
| Second step | 181–240 | 6.2 | 3.91 | 1.39 | 0.9 |
| Third step | 240 | 8 | 5.71 | 1.39 | 0.9 |

[1] Data Source: Tianjin Development and Reform Commission, http://www.tjdpc.gov.cn/.

*5.4. Environmental Problems Produced by Concentrated Brine*

Environmental problems produced by concentrated brine has gained attention to many researchers [15,34]. The desalination capacity of China will increase to 2.5 million–3 million cubic metres per day by 2020, according to the "Special Plan for Seawater Utilization". Taking the RO method as an example, this figure means that nearly 1,500,000 $m^3$ of concentrated brine will be produced per day in the future. Retentate brine has a high salt content and temperature, and it also contains chemical substances introduced by the pretreatment of seawater. In Jiaozhou Bay, the seawater exchange period is approximately 60 days [95]. If 200,000 $m^3$/day retentate brine is discharged into Jiaozhou Bay, the average salinity will rise by approximately 0.3 salinity units every year. After 30 years, the average salinity will be more than 4% [96]. For seawater desalination by the thermal process, retentate brines with high temperatures will cause the local seawater temperature to rise, affecting the physical properties of seawater. Directly or indirectly, this addition will lead to deterioration in the quality of seawater. A variety of chemicals are added during the pretreatment performed before desalination, as well as during the process of desalination itself. These chemicals include fungicides, coagulation agents, scale inhibitors, corrosion inhibitors, defoaming agents and reducing agents. The desalination devices also need to be cleaned regularly using acids and alkalis. These agents will be introduced into the ocean with the concentrated brine, and they will have some effect on the marine environment [15].

At present, the main method used to dispose of retentate brine is to discharge it into the sea. However, the ability of the sea to consume concentrated brine is limited [34]. Over the long term, the seawater near desalination projects will face ecological crises. Therefore, it is necessary for the byproducts of desalination to be used comprehensively as chemical resources. Using the typical "water-steam-water" material circulation and energy gradient, the Shougang Jingtang steel plant has gone furthest in achieving the reasonable use of energy in desalination. The seawater satisfies the demand for high-quality cooling water in ironmaking, steelmaking and the production of rolled steel. Retentate brine is made into solid salt in saltworks. Through industrial and agricultural consumption, it will be put into the sea eventually. This circular and efficient procedure enables reaching the goals

of maximizing energy production and minimizing pollution production. As a successful application of the thermal desalination process in the iron and steel industry, the Shougang Jingtang steel plant provides the missing piece in the use of desalination in domestic steel plants, and it provides a solution to the lack of water resources available to the iron and steel industry.

*5.5. Lack of Competitiveness and Complete Supporting Policies*

Zou et al. analyzed the economic effects of seawater desalination in China, and the results showed that the national economy has obvious demand-driven and supply-driven effects [97]. The cost of producing desalinated seawater is approximately 7–8 RMB/m$^3$. This total cost is much higher than the cost of water from conventional sources and its distribution cost. The present situation of seawater desalination is in accordance with the market-oriented operation, and it lacks complete government financial support and price subsidies. The selling price drops relative to the purchasing price, which restricts the development of this industry. Desalinated seawater from the Qingdao project is introduced into the municipal network. In that case, the desalination cost is 5.45 RMB/m$^3$. The government has agreed to purchase this water at a cost of 6 RMB, which is far higher than the current domestic water price of 1.8 RMB. More than ten years will be needed to recover the cost. Based on a capacity of 100,000 m$^3$ per day, the government needs to pay approximately $1.53 \times 10^8$ RMB every year to make up the payment deficiency.

On the other hand, China attaches great importance to the construction of water conservancy facilities. Public infrastructure, reservoir building, river management and water diversion projects are financed by the government or by other financial support. These subsidies exist for a long time, and the amount is increasing year by year, which reduces the cost of water production from conventional sources [89]. In addition, conventional water sources provide water on a large scale, diluting the operating cost. Compared with conventional water sources, seawater desalination, which is a new type of water supply, is not dominant in terms of policy support and economic scale. It has difficulties in competing with conventional water sources.

To alleviate the lack of competitiveness of seawater desalination and meet the need for supporting policies, the following suggestions are introduced.

1.  A scientific formulation of the desalinated water price will be taken as the fundamental basis of government subsidies to desalination plants and water-consuming enterprises. The government should encourage production-oriented enterprises and enterprises that consume large amounts of water to use desalinated seawater preferentially, providing preferential policies for the companies to help them recover their losses on water costs. For the desalinated seawater supplied to municipal networks, water supply enterprises should sign long-term procurement contracts with desalination plants and charge the desalination plants according to the actual amount of desalinated seawater in each municipal network. The water price difference between the market price and the purchase price should be compensated by the government.

2.  Developing the demand for desalination in the production field. The price of industrial water in the coastal areas of China is generally higher than that of domestic water. The desalination cost is close to the industrial water price in some cities with water shortages, which increases the application prospects of desalination in industry. By strengthening the policy guidance, more enterprises, especially those that consume large amounts of water, can be induced to use desalinated seawater. The Ministry of Water Resources issued the "Plan for the Reform and Development of Water Conservancy in the 13th Five-Year Period" in December 2016. It proposed that the government should encourage coastal areas and industrial enterprises to carry out demonstration projects using seawater. Desalinated seawater should be used preferentially in the industrial enterprises where it can be applied.

*5.6. Lack of Perfect Laws, Regulations and Standards*

As Zhu et al. [15] mentioned, at present, seawater desalination does not have the laws and regulations needed to regulate its construction, development and management in China, the development of seawater utilization work, in accordance with National 13th Five-Year Plan for Seawater Utilization. The orientation, supervision, responsibility, power and interest of desalination projects are not well defined. Research into the safety of using desalinated seawater is still lacking. For example, Tianjin continues to use the "Domestic Drinking Water Health Standard (GB5749-2006)" as its desalinated seawater quality standard. This standard specifies the maximum concentrations of poisonous and harmful substances but it does not define the lower limit values of ion concentrations, RO water is typically alkaline and needs chemicals to be added to allow it to be drinkable and not corrode pipes, otherwise, it will influence the stability of the pipe network. Lower limit values of ion concentrations also influence public health, and some ions are significant when the desalinated water is used for irrigation (e.g., upper threshold for boron).

To encourage the development of relevant laws, regulations and standards, the following countermeasures are presented:

1.  In establishing a system for administering desalination projects, the strategic positioning of desalination should be clarified in the law to promote its standardized development. In addition, the Department of State, including the Ministry of Water Resources, the State Oceanic Administration, the Ministry of Environmental Protection, and other local departments, as well as associations involved in marine management, need to confirm their respective functions, responsibilities, rights and interest distributions, and so on. Currently, the Ministry of Water Resources has a Department of Unconventional Water Resources, which is responsible for the administration of unconventional water sources and overall planning specifically including the use of seawater. In addition, the "Plan for the Reform and Development of Water Conservation in the 13th Five-Year Period", which was issued in December 2016, noted that unconventional resources should be incorporated into the regionally unified allocation of water resources. As an important means of realizing water conservation priorities and system management, strengthening the development and utilization of unconventional water resources is of great significance to ease the contradiction between water resource supply and demand in China.

2.  We should determine whether the quality of desalinated seawater is satisfactory for transport within municipal pipe networks, establish quality standards for desalinated seawater, develop a mineralization processing standard for desalination, and build a system for determining the appropriate scales of new projects. At present, sulfuric acid is mainly used to dissolve limestone to mineralize desalinated seawater in China [98]. The development of the desalination industry is expected to be guided and standardized in such aspects as the development of resources, environmental protection, safety and industrial development. We also should perform research on desalination and draining, raw materials and reagents, desalination technology, testing technology, engineering designs, operation management practices, desalination regulatory standards and the design of related equipment and quality standards, as well as other topics [99]. These developments will strengthen the guidance available to the desalination industry.

## 6. Concluding Remarks

This paper introduces comprehensive seawater utilization in China in three aspects, including the desalination of seawater, the direct use of seawater, and the use of seawater as a chemical resource.

First, the history of development and the status quo of desalination in China were analysed. Since the "7th Five-Year Period" (1986–1990), China has supported research into and development of related technologies. By the end of 2017, the available desalination capacity in China had reached $1.18 \times 10^6$ m$^3$/day. China had built 136 desalination engineering projects. These projects are mainly concentrated in Tianjin, Shandong, Zhejiang, Hebei and Guangdong Provinces. The major technologies

used to perform seawater desalination in China, reverse osmosis (RO), low-temperature multi-effect distillation (MED) and multi-stage flash distillation (MSF), have further broadened the desalination market. By the end of 2017, 117 plants employ RO technology to desalinize seawater, whereas 16 plants employ the MED technique.

Next, the direct utilization of seawater was discussed. By the end of 2017, the amount of seawater used for cooling water had reached $1.34 \times 10^{11}$ m$^3$. In 2017, the annual seawater utilization in Guangdong, Zhenjiang, Fujian, Liaoning Provinces were $4.18 \times 10^{10}$ m$^3$, $3.07 \times 10^{10}$ m$^3$, $2.26 \times 10^{10}$ m$^3$ and $0.92 \times 10^{10}$ m$^3$, respectively. Furthermore, by analysing the comprehensive utilization of seawater in China, the perspective of optimizing the utilization of seawater resources was presented to achieve the "minimum quantitative" discharge.

Finally, restrictive factors and potential countermeasures of the increased use of seawater desalination are investigated. Several specific recommendations are presented, specifically improving the laws, implementing regulations and standards related to desalination, strengthening the policies that support the enterprises that use desalination, improving the localization rate of key technologies and equipment gradually and devoting additional attention to the problems associated with brine processing.

The utilization of seawater resources is in the developmental stage in China, and there are differences in the utilization of water resources in different regions. Improving the relevant support policies for desalination is necessary. The preferential policies currently introduced have not been sufficiently supportive for desalination enterprises, and there is a lack of specific support, guidance and encouragement policies such as running water and South-to-North Water Transfer. The seawater desalination price mechanism is still unreasonable. At present, a benign water price mechanism has not been established, which has caused seawater desalination to remain at a disadvantage in terms of price. The strategic positioning of seawater use is not yet clear. In the Water Law of the People's Republic of China (revised in July 2016), seawater was not included in the conventional water resources management system, and the corresponding preferential policies could not be obtained. In future law formulation, consideration should be given to increase the utilization of seawater. Compared with Freshwater resources, seawater does not have a prominent advantage in terms of price. Along with the severe shortage of water resources, long-term work is still required to reasonably formulate and improve regulations and standards, and to determine how to incorporate seawater into a regional unified allocation of water resources. These issues require further comprehensive study.

**Author Contributions:** The article was mainly written by S.G., Q.B. H.W., C.W. and Z.Z. provided many valuable comments. All authors read and approved the final version of the manuscript.

**Funding:** This study was supported by the National Natural Science Foundation of China (Grant No. 51879010, 51379006, 51479003) and the 111 Project (Grant No. B18006).

**Acknowledgments:** The authors would like to thank all the anonymous reviewers and editors for their valuable comments and constructive suggestions, which led to the improvement of the presentation of this paper.

**Conflicts of Interest:** The authors declare no conflicts of interest. The founding sponsors had no role in the design of the study; in the collection, analysis, or interpretation of data; in the writing of the manuscript; or in the decision to publish the results.

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
