# Peer review of "Comprehensive Utilization of Seawater in China: A Description of the Present Situation, Restrictive Factors and Potential Countermeasures"

_water, doi:10.3390/w11020397_

Round 1

Reviewer 1 Report

Generally speaking, the paper is interesting and sheds light on an important subject. However,  it can be classified as a report, rather than a scientific paper.

Specific comments:

Abstract:

Line 20: I recommend replacing "present status (2015)" with "recent status (2015)"

Introduction

1.       Lines 38-39: You mention the use of seawater in several application. Is it also used for hydraulic fracturing?

2.       Please add reference of date to the data regarding the total desalination capacity (line 42)

3.       Lines 61-62: the sentence " except in Perth, due to the end of a drought" is not clear. Please explain.

4.       Page 2: in general, you mention the environmental issues related to desalination, but you overlook the issues related seawater intake, which has an adverse effect on plankton population, for example.

5.       Lines 99-103: the units of the data are not clear. Are these annual demands? Annual utilization?

6.     Lines 110-111: in what respect the relevant studies are insufficient?

Regarding the data given in the paper related to china, some comparison to worldwide data could make the paper more interesting. For example, is direct use of seawater for domestic flushing popular in other places?

Seawater desalination in china

1.     Lines 225: what do you mean by double membrane RO?

Direct utilization of seawater in China

1.       What about the use for aquaculture of sea fish? Algea?

2.       Figure 10: capital letter for Shandong

3.       Lines 306-318: what about environmental issues? For example, what is the faith of the seawater after used for flushing? How does it affect the wastewater composition and treatment?

4.       Lines 324-328: long term irrigation with slightly elevated TDS water was observed to adversely effect the soil. See for example the case in Israel, where replenished effluents were used for irrigation for long periods. Please refer to this issue in china too.

Utilization of seawater as a chemical resource and structural optimization in China

1.       The term "strong brine" is not common. Please use simply "brine" of "retentate".

2.       Line 355: magnesium is often also extracted from seawater.

3.       Line 369: I think you have a mistake here: in seawater RO plants the brine salinity is ~70,000 mg/l, not 20000 mg/L.  

4.       Line 371: for what kind of use do you mean here: " he concentrated remaining brine can also be used comprehensively

Restrictive factors and potential countermeasures for

1.       Line 465: I think the figure you refer to is missing.

2.       Line 507: specify the units of the capacity

3.       Line 602: the salinity of the dead see is around 35%, not 4%.

4.       Line 671: lower limit values of ion concentrations influences public health too (e.g. Ca2+, Mg2+, I-), and some ions are significant when the desalinated water are used for irrigation (e.g. upper threshold for boron).

5.       Paragraph starting in line 689: please specify which remineralization methods are currently applied in China?

Author Response

Thank you very much for the valuable comments of the reviewers. We have already answered each question. See the word document for details.

Reviewer 2 Report

The manuscript (MS) topic deals with the utilization of seawater in China, which is a very broad topic and consequently difficult to delimit. It is mainly focus on the production of desalinated seawater, although also includes a section about the direct use of seawater (industrial cooling in coastal areas and other minor uses), as well as another section about the utilization of seawater as a chemical resource (mainly salt production for NaOH and Na2CO3 industries). The MS topic falls into the journal scope, but my recommendation is that the MS should not be accepted for publication in WATER journal for the following reasons:

-        The main reason is that about 70% of the MS content has already been published by two of the authors (Zhongfan Zhu and Hongrui Wang) in another research journal: Zhongfan Zhu,  Dingzhi Peng,  Hongrui Wang. Seawater desalination in China: An overview. Journal of Water Reuse and Desalination, In Press (https://doi.org/10.2166/wrd.2018.034). In my opinion, there is a clear parallelism between the content of both MS, which manage just the same information:

“The course of desalination development” in WATER has the same information than “History of seawater desalination” in JOURNAL OF WATER REUSE AND DESALINATION.

“The status quo of desalination development” in WATER has the same information than “Present condition of utilization of seawater desalination” plus “Geographic distribution” in JOURNAL OF WATER REUSE AND DESALINATION.

Table 1 in WATER has the same data as section “History of seawater desalination” in JOURNAL OF WATER REUSE AND DESALINATION.

Figure 4 in WATER  has the same data as Figure 1 in JOURNAL OF WATER REUSE AND DESALINATION. 

Figure 3 in WATER  has the same data as Figure 2 in JOURNAL OF WATER REUSE AND DESALINATION. 

Figure 5 in WATER has the same data as Figure 4 in JOURNAL OF WATER REUSE AND DESALINATION. 

Figure 7 in WATER has the same data as Figure 3 in JOURNAL OF WATER REUSE AND DESALINATION.

Table 3 in WATER has the same data as Table 1 in JOURNAL OF WATER REUSE AND DESALINATION.

Figure 7 in WATER has the same data as Figure 3 in JOURNAL OF WATER REUSE AND DESALINATION.

“Restrictive factors and potential countermeasures for the comprehensive utilization of desalinated seawater in China” in WATER present a discussion similar to that in “Existing problems and suggestions regarding seawater desalination” in JOURNAL OF WATER REUSE AND DESALINATION.

-     WATER publishes original research papers, critical reviews and short communications. The MS is not a research article since methodical and/or experimental details are not presented. Besides, it does not follow the usual structure in research papers (Introduction - Materials & Methods -Results & Discussion – Conclusions). The MS is not a short communication (more than 12.000 words cannot be “short”). In my opinion, the MS aims to be a critical review focus on local aspects (China) about the utilization of seawater, which is a too broad topic for a critical review. The handled data ends in 2015, so it is not updated as it should be required in a review. Therefore, the MS neither can be considered a critical review, it looks more like technical report for any other purpose.

-        The original parts of the MS (direct use of seawater and the utilization of seawater as a chemical resource) are poorly analysed and are not integrated into the other common sections of the MS (Introduction, Discussion, and Conclusions). These parts are like an addition to justify that the MS is not just the same as “Seawater desalination in China: An overview” in Journal of Water Reuse and Desalination, and they have not enough entity to be published.

-        The MS is unnecessarily long and difficult to read for an international reader. The authors should try to stick to the core findings. Some figures and tables are redundant (e.g. Table 3 and Fig. 6) or unnecessary (e.g. Fig 1 and Fig. 13)

Author Response

Thank you very much for the valuable comments of the reviewers. We have already answered each question. See the word file for details.

Reviewer 3 Report

Review Report 1

Manuscript ID: water-432667

Comprehensive Utilization of Seawater in China: A Description of the Present Situation, Restrictive Factors and Potential Countermeasures

Comments

1.       Line 20 – Why is the present status 2015? Today is 4 years on. Change wording to reflect the fact that you are basing your study on historical 2015 data

2.       Line 32/33 – Is this correct– references required for your point.

3.       Line 35-40 – No problem with the points, but you should give references for each approach.

4.       Line 42 – replace is currently with in 2015 and provide a reference for the 51 million. It is my understanding that At the end of 2015, there were approximately 18,000 desalination plants worldwide, with a total installed production capacity of 86.55 million m3/day or 22,870 million gallons per day (MGD). Around 44% of this capacity (37.32 million m3/day  or 9,860 MGD) is located in the Middle East and North Africa. While desalination in that region is projected to grow continuously at a rate of 7 to 9 percent per year, the “hot spots” for accelerated desalination development over the next decade are expected to be Asia, the US and Latin America. http://www.iwa-network.org/desalination-past-present-future/

5.       Line 43 – This list is not correct. I refer you to https://www.researchgate.net/figure/Global-installed-desalination-capacity-2010-2016-Adapted-from-11_fig2_272756219

6.       Line 47 reference required – a reference is required for each point made unless it is a new point made by yourselves

7.       Line 48-62 reference required for each point made

8.       Line 63 –define what the fresh water resources per capita are for China and note why you think that this indicates a shortage and supply appropriate references foor the points made

9.       Line 64/65 – points made are OK but wording is unclear – expand what you mean to improve readability and supply references/data for the points made.

10.    Line 66 – name the 11 provinces at this point or show them on a figure – to improve reader clarity

11.    Figure 1b – What do the volumes refer to m3 for area with colour or m3/km or something different – please note on Figure

12.    Figure 1 caption – please note reference sources for the information in the figures (a) to (d)

13.    Line 80 – reference required

14.    Line 82/83 – References required for the point made. Is the mature technology you are referring to actually Chinese or is it imported technology or a combination?

15.    Please place a list of abbreviations used at the start of the MS

16.    Line 86 – MED and RO technologies are not Chinese in origin but were developed elsewhere. I accept that there are Chinese companies that do supply these technologies.

17.    Line 111 change “are insufficient” to “are in our view insufficient” I note that there are about 130,000 papers relating to desalination and China of which 32,000 papers address seawater desalination in China. Please clarify for the reader why you think that this research base is insufficient and in which areas is more research required in your view. Parts of the introduction have a very similar wording to the paper by two of the authors accessed at 

https://iwaponline.com/jwrd/article/doi/10.2166/wrd.2018.034/64192/Seawater-desalination-in-China-An-overview and include many of the same errors the earlier paper contains. Table 1 of this paper (a version of Table 3 in this MS) should be merged into Table 1 of the current MS (together with source references) to provide the reader with the advances between 2005 and 2015. Having read both papers I think that Fig2 of this paper should be supplemented in the text by a version of Table 2 in your earlier paper to improve readability and improve reader access to the policy issues you are trying to address

18.    Line 123 - 136 – references required to support each statement made. These supporting references can be in Chinese or the relevant local language

19.    Figure 2 – source references required. In the text on the figure please change “The opinion” to something else as your wording makes no sense. Do you mean consultation?

20.    Table 1 – references required for each stage and project noted. These can be references to newspaper/magazine articles or government publications.

21.    Line 141-157 – references required for each point made

22.    Line 157 – is the unit actually 140 K m3/second (i.e. 201 million m3/day) as written, or something different, and is the date actually 2015 or something different, 1996-2006 cannot be correct.

23.    Line 160-164 – poor English, please reword.

24.    Line 160 – 169 reference required for each point made

25.    Table 2 – reference required for each technology showing Chinese application

26.    Line 175 replace metres with metres/day

27.    Line 177-697 – references required for each point made/plant identified

28.    Figure 6 – data source references required

29.    Line 252 – Is this only in China, please be specific

30.    The English throughout is generally good, though in my view many sentences and paragraphs in the text could be made more concise and focussed to improve readability. It would make reading easier if you could shorten each sentence to make single focussed point, and where a single sentence makes numerous points it should be split into a number of shorter concise focussed sentences where each sentence makes a single point – i.e. a style issue

31.    The authors have used 2015 as a cut off year and have therefore ignored the advances in forward osmosis and capacitive osmosis. They have also ignored the vast steps made in Chinese agriculture in terms of increasing salt tolerance of major food crops, the use of diluted saline water for irrigation, etc.

32.    The study is a high level policy review. It covers the main technologies used in China but needs to be made more concise and focussed. This could be helped by the use of more sub headings

33.    The various economic issues raised in https://www.technologyreview.com/s/601861/chinas-massive-effort-to-purify-seawater-is-drying-up/ are not addressed. This is surprising since they will define whether the technology expansion stalls.

34.       The SWRO projects which highlight groundbreaking Chinese SWRO technology such as A pilot study of UF pretreatment without any chemicals for SWRO desalination in ChinaDesalination 207 (2007) 216–226 are ignored

35.    The groundbreaking Chinese work on sea ice desalination technology is ignored Study on sea ice desalination technology  Desalination Volume 245, Issues 1–3, 15 September 2009, Pages 146-154

36.    Chinese work on VMD desalination is ignore Desalination of oil-field wastewater via vacuum membrane distillation Mem Sci Technol 2004 (1)

37.    Why was Economic effects analysis of seawater desalination in China with input–output technology Desalination 380 (2016) 18-28 ignored

38.   I think that you should increase the review section of your study to actually document Chinese research and projects associated with desalination to 2015. You should consult Google Scholar, Scopus/other academic data bases and the Chinese Patent Office Data base. The paper as it is written is really very superficial in detail and reads more like a high level government report, without reference support, than a scientific policy paper. In many places you quote from a China Desalination Yearbook, and indeed the reader is led to believe that you have copied Figures 3-5, 7,8 from this yearbook and Figures 9,10 from another government report. Some Figures such as Fig 12 have no source references (please supply). This is a useful Figure and I think that the MS (and its policy implications for planners) would be significantly improved if you could construct and add similar figures for the various desalination technologies and applications noted in the text. I have not commented on your interpretation of government policy or your recommendations, as in this type of paper the same data set is open to many alternative interpretations and visions.

39.   The supplementary material file is not relevant to this study. Please delete

Author Response

(The authors gave the same response as above.)

Round 2

Reviewer 1 Report

the manuscript was sufficiently improved.

Reviewer 2 Report

The authors have revised the manuscript and have addressed the questions I raised. The technical content in this revised version and its presentation has been improved and the detected problems have been partially solved and/or clarified.

In my opinion the argumentation and discussion of the manuscript has also been improved and it can be published.

Reviewer 3 Report

I have gone through the revised MS and your comments. There are a few typos in the MS but these can be addressed at the proof stage (.e.g. Line 85). I have no further comments on the MS